# Pocketable Labs for Everyone: Synchronized Multi-Sensor Data Streaming and Recording on Smartphones with the Lab Streaming Layer

**DOI:** 10.3390/s21238135

**Published:** 2021-12-05

**Authors:** Sarah Blum, Daniel Hölle, Martin Georg Bleichner, Stefan Debener

**Affiliations:** 1Neuropsychology Lab, Department of Psychology, University of Oldenburg, 26111 Oldenburg, Germany; stefan.debener@uol.de; 2Cluster of Excellence Hearing4all, 26111 Oldenburg, Germany; 3Neurophysiology of Everyday Life Group, Department of Psychology, University of Oldenburg, 26111 Oldenburg, Germany; daniel.hoelle@uol.de (D.H.); martin.georg.bleichner@uol.de (M.G.B.)

**Keywords:** mobile computing, sensor integration, mobile health, smartphone sensor integration, time series analysis

## Abstract

The streaming and recording of smartphone sensor signals is desirable for mHealth, telemedicine, environmental monitoring and other applications. Time series data gathered in these fields typically benefit from the time-synchronized integration of different sensor signals. However, solutions required for this synchronization are mostly available for stationary setups. We hope to contribute to the important emerging field of portable data acquisition by presenting open-source Android applications both for the synchronized streaming (Send-a) and recording (Record-a) of multiple sensor data streams. We validate the applications in terms of functionality, flexibility and precision in fully mobile setups and in hybrid setups combining mobile and desktop hardware. Our results show that the fully mobile solution is equivalent to well-established desktop versions. With the streaming application Send-a and the recording application Record-a, purely smartphone-based setups for mobile research and personal health settings can be realized on off-the-shelf Android devices.

## 1. Introduction

Smartphones have become relevant in various research disciplines and mobile health applications, and due to their portability, they enable the investigation of previously uncharted questions [1,2,3,4]. They can be used for capturing brain activity in ambulatory settings [5,6,7,8,9,10,11], sleep patterns outside of sleep laboratories [12,13], subjective experience in various contexts [14,15,16] and body activity while moving freely [17,18,19,20,21,22,23]. While less powerful than full PC desktop systems, smartphones offer sufficient computational power and a multitude of sensors to serve as standalone research instruments [6,21,24,25,26]. Often, these built-in sensors are combined with body-worn sensors to capture whole-body movements as well as human–environment interactions in great detail, thereby opening the door to various mHealth applications [27,28,29,30,31]. To mention a few examples, the accurate recording of muscle (measuring electromyogram, EMG), heart (electrocardiogram, EKG) and brain activity patterns (electroencephalogram, EEG) in mobile setups merge portable systems such as smartphone sensors with complementary, body-worn sensors [8,12,32,33,34,35].

Independent of the application or hardware used, all combined sensor data from different devices require precise temporal synchronization for relating information captured in different modalities to each other correctly [30,32,33,36,37,38,39,40,41].

This is not easily achieved and remains a challenge in many applications. Each sensor device typically time stamps data based on its respective internal clock. The relation to timestamps from other devices or additional sensors is inherently unknown, and over time, the clocks of different devices will likely diverge. As a consequence, the resulting data streams cannot be related to each other directly, and an interpretation becomes impossible. For stationary systems and smartphones alike, the initial synchronization of clocks can be realized in two ways [42,43]. In a network-based environment, a master clock can be defined and can be used to determine the initial differences between the master and every other clock. Assuming these differences remain stable over time, the matching timestamps of all clocks can then be determined. In non-network setups, the timestamps of every device can be related to each other by recording a sharp impulse (i.e., synchronization reference event) on all devices at the same time, effectively serving as a reference point for all clocks (as outlined in [36,44,45]). This typically requires wiring up the different devices.

However, initial synchronization alone is typically not sufficient as clocks are likely to diverge over time due to distinct paces between them [36]. Instead, the initial synchronization procedure needs to be repeated at least once to determine the drift of clocks over time. Synchronization intervals can be large in some cases, but in the case of strongly diverging clocks, more frequent synchronization events are necessary. The implementation of repeated synchronization events can be achieved with a software agent running on each device or as a cloud service [46], without the use of wires and connectors.

For this purpose, the Lab Streaming Layer (LSL) has gained popularity in the field of multi-modal data recording in recent years [31,47,48]. LSL (RRID:SCR 017631, https://github.com/sccn/labstreaminglayer (accessed on 16 November 2021)) is an open framework consisting of a core library and interfaces for many common programming languages, thus enabling the usage of LSL on a wide range of devices. It uses standard network protocols to send and receive data streams over the local network. All LSL data streams are available with initial synchronized clocks and known drift behavior between devices over time. By default, clock synchronization is performed every 5 s and data from different sensor streams are stored along with all information needed for time synchronization, using a well-defined, standardized file format (extensible data format, xdf, https://github.com/sccn/xdf, accessed on 1 October 2021). For recording LSL data on desktop systems, the LabRecorder software (https://github.com/labstreaminglayer/App-LabRecorder, accessed on 1 October 2021) was developed a few years back, but no versatile, open-source LSL recording application for mobile devices has been published yet. This necessity to use stationary desktop devices has hindered mobile and ambulatory research.

Therefore, we set out to fill this gap. The functionality of the LSL LabRecorder is now available on Android in an application we call Record-a. In order to present the flexibility of this new solution, we present three Android applications. One application turns smartphone sensor data into LSL data streams: Send-a. A second application streams artificial signals on multiple channels: Sine-Wave app. All apps can be used independently and finally extend the LSL streaming and recording framework to mobile application scenarios. We describe, evaluate and discuss Send-a, Sine-Wave app and Record-a in scenarios that are relevant for, but not limited to, mobile multi-sensor setups, mhealth and clinical research.

### 1.1. The Smartphone Apps: Record-a, Send-a and Sine-Wave App

The Record-a, Send-a and Sine-Wave apps are Android applications written in Java using the Java Native Interface to call functions from the LSL core library, written in C++ (https://docs.oracle.com/javase/8/docs/technotes/guides/jni/spec/jniTOC.html, accessed on 3 August 2021). Both the LSL library functions, and our apps, are open-source projects (all scripts, apks and source code available here: https://github.com/s4rify/Pocketable-Labs (accessed on 16 November 2021), https://github.com/sccn/labstreaminglayer (accessed on 16 November 2021). All apps call the LSL library functions so that data and stream handling, as well as storage in standardized file formats, was directly realized using the LSL methods, matching the desktop equivalents as closely as possible.

Both Send-a and Record-a were developed to run as foreground services (https://developer.android.com/guide/components/foreground-services, accessed on 24 September 2021). This prevents the operating system from shutting them down or limiting their processing power, even if they are currently not in the foreground (that is, visible to the user and in full screen).

Our apps can be used in a variety of setups, either on a single device or in combination with other devices. They can run on the same or different Android smartphones and tablets without the need to alter the operating system or change the device itself. For all evaluations described in this report, we used the apps on physical smartphones available to us, these were:Huawei Honor View 10 (Android 10, round corners in all figures)Samsung A51 (Android 11, square corners in all figures)

### 1.2. Record-a

Record-a uses the LSL library to find all streams visible in the network and record them into xdf files. The implementation of Record-a follows the recommendations of the LSL developer community (https://labstreaminglayer.readthedocs.io/dev/app_dev.html, accessed on 1 July 2021; https://github.com/labstreaminglayer/liblsl-Android/tree/master/AndroidStudio accessed on 14 July 2021). Record-a is ignorant of the type, sampling rate or origin of streams.

Record-a will scan the network on startup and initially select all detected streams for recording. This list of streams is presented to the user in the GUI who can choose the streams to be recorded. Additionally, users can specify the name of the recording, which is always appended with the current time and date information from the device to ensure a unique file name to prevent the accidental overriding of an existing file. After the final selection of streams and initiation of the recording, Record-a will inform the user about the beginning of the recording and whether it is able to safely run in the background on this device. Next, Record-a will write the header information from every stream that is selected in the file following the xdf specifications. During recording, Record-a stores data from all streams for 500 ms in internal buffers and then writes these segments to the file iteratively. Every sample written to file is safe and preserved even if the app is stopped unexpectedly, e.g., due to a dying battery. Iterative recording from internal buffers to the file ensures that a maximum of 500 ms of data can be lost. In addition to the data, Record-a will store information for the synchronization of all data streams in the form of clock offsets computed relative to a master clock, in this case, the clock of the device on which Record-a is running. These clock offset values are recorded every 5 s and are additionally stored in the file footer, thereby matching the clock handling of LabRecorder on a desktop. The footer also contains information about the absolute sample count that was received for each data stream to compute the effective sampling rate and the first and last timestamp of the recording. Access to the xdf file is shared among different stream recordings so that the absolute number of streams to record from is not limited by the file handling routine. Error handling and stability are described in more detail for different scenarios in Section 3.5.

### 1.3. Send-a

The second LSL mobile app Send-a is also written in Java and calls the LSL functions using the Java Native Interface. Send-a pulls values directly from the device sensors using Android’s sensor manager class (https://developer.android.com/reference/android/hardware/SensorManager, accessed on 24 September 2021). Sensor values are stored in small internal buffers of one sample per sensor and then streamed as LSL streams. Available sensors depend on the smartphone model. From a software perspective, Android generally supports three categories of sensors: motion sensors (accelerometer, gyroscope), environmental sensors (temperature, light, gravity, proximity) and position sensors (GPS). Broadly speaking, the microphone can also be seen as a sensor and is generally supported by Android and Send-a (https://developer.android.com/guide/topics/sensors/sensors_overview, accessed on 28 September 2021). In some devices, the camera can be accessed in a similar way. In its current state, Send-a does not stream video or GPS data.

Some sensors are realized as hardware sensors. They are physical components that measure environmental influences such as acceleration, rotation or light intensity directly. These sensors show the same behavior on all devices that have them. Some sensors are realized as software sensors that derive data from hardware sensors and provide these as interpreted values. Software sensors, therefore, mimic hardware sensors, but they operate in deducing their measurements more indirectly, for instance, by providing a step count deduced from a combination of several sensor values. The interpretation, as well as the precise computation to deduce these interpreted measurements, can vary between manufacturers or devices (https://developer.android.com/guide/topics/sensors/sensors_overview, accessed on 28 September 2021).

Send-a detects sensors of the device and creates LSL data streams for every sensor the user selected in a list of available sensors. Send-a registers a listener for every sensor, which is notified by the operating system whenever the sensor manager registers a change in sensor readings. This event is also triggered when the operating system changes the sampling rate of the sensor. Importantly, the rate at which the sensor manager pulls new values from the sensors is variable and Android does not offer to define a constant sampling rate. This behavior is independent of the usage of LSL; it is a property of Android’s sensor management. Every sampling interval specified by an application is treated only as a suggestion to the operating system, which can result in unexpected behaviur from a user’s perspective. This behaviur is documented both for hardware and software sensors (https://developer.android.com/guide/topics/sensors/sensors_overview, accessed on 28 September 2021).

Therefore, Send-a does not allow the user to specify the sampling rate manually. Instead, it implements a thread handling routine which asks the sensors for new values with a constant, albeit rather long, delay. In its current version, this delay results in an effective sampling rate of about 100 Hz, but a large processing load will lead to a reduced sampling rate for all streams. Notably, the sampling rates are then decreased for all sensors alike, resulting in the same effective sampling rate for all sensors except the microphone. The effective (resulting) sampling rate is stored in the xdf file together with the intended sampling rate so that the data can be imported with correct sampling information. It was previously mentioned that the microphone is treated differently from other sensors by the OS; this also applies to the sampling rate stability. The sampling rate of the microphone can either be set to 44 kHz or 48 kHz. The OS will make sure that this rate is kept constant. It will never reduce or elevate this sampling rate.

### 1.4. Sine-Wave App

Sine-Wave app is a simple data generation app that streams sine wave signals on ten channels with a sampling rate of 250 Hz. Each channel contains a sine wave signal with a different frequency. As such, the signal generated by the Sine-Wave app is similar in data size and form to neural signals provided by commercially available setups. This app was mainly developed to simulate a scenario in which the system load is lower than Send-a since the Sine-Wave app does not access any system resources.

## 2. Methods

We evaluated our apps in three scenarios and a timing test. Our scenarios were chosen with the aim to show that a combination of Record-a and stream sources can be used either on a single smartphone alone, on a combination of multiple smartphones, and in a network of mobile and stationary devices.

### 2.1. Scenario 1: Data Streamed from Android Devices Recorded on a PC

In many data collection setups, different sensor readings originating from various devices are combined and centrally recorded on a stationary PC. In this scenario, we show how our apps can be an integral part of such a setup. In these setups, the combination of integrated smartphone sensors with stationary systems might be a powerful addition to the investigation of diverse research questions. In this scenario, data were sent out from two physical smartphones to a PC. On the phones, either the Sine-Wave app (Scenario 1A) or Send-a (Scenario 1B) was running.

### 2.2. Scenario 1A: Sine Wave Data Streamed from Android Devices, Recorded on a PC

Sine-Wave app was running on two phones (Huawei Honor View 10 and Samsung Galaxy A51), streaming sine waves with different frequencies on ten channels with a nominal sampling rate of 250 Hz. On a PC, LabRecorder (version 1.13.0-b13, https://github.com/labstreaminglayer/App-LabRecorder, accessed on 1 October 2021) recorded the data from both phones. All devices were connected to the same WiFi network.

### 2.3. Scenario 1B: Sensor Data Streamed from Android Devices, Recorded on a PC

In this variant of the same scenario, Send-a was running on the two phones. Send-a streamed sensor values of rotation, gravity and accelerometer sensors. On a PC, the LabRecorder recorded all streams into one file. Both phones were moved together while being held display-up on top of each other. The movements were exaggerated movements of the devices in space, walking and jumping while holding the phones in both hands.

### 2.4. Scenario 2: Data Streamed from PC, Recorded on an Android Device

In this scenario, a laptop sent out a stream in the form of neural data (8 channels, 250 Hz sampling rate) generated in MATLAB with a custom script (version 2020a, part of the repository for this paper: https://github.com/s4rify/Pocketable-Labs/blob/master/Shared_Functions/send_eeg_and_markers.m, accessed on 16 November 2021). These data were then recorded with Record-a on a smartphone and LabRecorder on Windows (same PC that sent out the stream) simultaneously. All devices were connected to the same WiFi network.

### 2.5. Scenario 3: Sensor Data Streamed from Android Device, Recorded on the Same Android Device

In this scenario, both Send-a and Record-a were running on the same physical Android device. In a long recording (30 min), the phone was kept lying on the desk while all sensors except the microphone were recorded. All data were captured from the internal sensors in the phone, data streams were sent out by Send-a and recorded by Record-a running on the same device. For validation purposes, the same data streams were also recorded by LabRecorder running on a PC connected to the same local network as the phone.

## 3. Results

Both data source apps, Send-a and Sine-Wave app, as well as Record-a, ran stable, without crashing once. We observed no data loss in the streaming setups. Sensor readings correlated highly between reference recordings using LabRecorder and Record-a. Additionally, Record-a successfully recorded data both from apps on the same device, a different mobile device and desktop devices in the same network.

### 3.1. Scenario 1A: Sine Wave Data Streamed from Android Devices, Recorded on a PC

The nominal sampling rate of the Sine-Wave app was set to 250 Hz, but since there is always additional load on the phone and the implementation was not focused on performance, we did not expect the sampling rate of 250 Hz to be reached. Accordingly, the effective sampling rates on our two phones were 203 Hz (Samsung) and 213 Hz (Huawei). We found the correlation of all channels between phones to be very high (>0.9).

Figure 1 show the setup on the left side of the figure. The right side shows two exemplary channels from the sine data. For the plot, data were imported without jitter handling and sampling rates between phones varied by 10 Hz. This deviation is shown here as diverging alignment between streams from phone one (blue line) and phone two (orange line). The streams briefly align and then diverge due to sampling rate differences. In the lower plot, both data streams were effectively resampled to a common sampling rate by plotting them on the same time scale.

### 3.2. Scenario 1B: Sensor Data Sreamed from Android Devices, Recorded on a PC

The nominal sampling rate for all sensors streamed by Send-a was set to 125 Hz in this evaluation. We observed different effective sampling rates due to the varying processing load of the system when evaluating a different number of sensors. When streaming one sensor only, the effective sampling rates were 114 Hz (Samsung) and 118 Hz (Huawei). When streaming three sensors, the effective sampling rates were 106 Hz (Samsung) and 116 Hz (Huawei). When streaming all sensors except the microphone, the effective sampling rates were 95 Hz (Samsung) and 111 Hz (Huawei). Figure 2 show the setup on the left side and data recordings of one software sensor (rotation) and two hardware sensors (gravity, accelerometer) on the right side. While the hardware sensor recordings correlated highly (0.95 and 0.98) despite measuring data on two different hardware devices, the software sensor showed more deviation in measured signals as indicated by a lower correlation between recordings (0.54).

### 3.3. Scenario 2: Data Streamed from PC, Recorded on an Android Device

The nominal sampling rate set in the MATLAB script was 100 Hz, the effective sampling rates recorded by the phone and the PC were both 94 Hz. This deviation of the sampling rate can be explained by MATLAB’s handling of the pause function, which usually pauses a bit longer than indicated. This is due to overhead of calling the function and scheduling resolution of the operating system (https://de.mathworks.com/help/matlab/ref/pause.html, accessed on 3 July 2021), thus explaining a lower effective sampling rate from the sender but displaying no difference in sampling rates on the receiving sides. Data from laptops and smartphones correlated with 0.96. An evaluation of the time difference between all samples indicated no data loss. Figure 3 show the setup on the left side and some exemplary data on the right side. Since the same data were recorded in both files, the blue plot showing data recorded by the PC and the orange plot showing data recorded on the phone overlap almost exactly.

### 3.4. Scenario 3: Sensor Data Streamed from an Android Device, Recorded on the Same Android Device

Effective sampling rates were recorded to be 76.5 Hz for all sensors on both devices. All sensors correlated highly (>0.9) between recordings indicating that data were recorded correctly by Record-a. Figure 4 show the setup on the left side of the plot and some recorded data on the right side. Since Record-a and LabRecorder recorded the same samples, data on all channels at every point in time was virtually the same.

### 3.5. Fault Tolerance and Stability: Record-a

In any regular recording case, the recording is stopped by the user. Record-a will write the footer information and then close the file. In the following section, we describe some potential error scenarios that demand different behavior from LSL recording software such as Record-a.

#### 3.5.1. Error Scenario: The Stream Outlet Pauses or Stops to Send New Samples

Record-a handles this situation the same way as LabRecorder for Windows and stops writing values into the file while waiting for the stream(s) to continue to send values. Once new samples are received, Record-a will continue writing the new values with their respective timestamps in the file. Both Record-a and LabRecorder resume recording, and the file will be closed in a regular way at the end of the recording. In case the stream does not resume, the recording continues without new values being received. The user can end the recording at any point, and the file will be closed regularly.

#### 3.5.2. Error Scenario: Record-a Is Terminated Unexpectedly

A different scenario could be that Record-a is killed during a recording, either because the device battery dies, or the user kills the app during recording. In both cases, the file will be closed and written to the disc. It will contain all samples recorded up to this point. Due to the interval with which Record-a writes values to file, a maximum of 500 ms of data can be lost. The footer will not be written; therefore, the clock-offsets for later synchronization of the recorded streams will be missing, but the file can be imported, and the data can be loaded. This behavior is the same as is implemented in LabRecorder for Windows.

## 4. Discussion

We present an open-source solution for completely mobile, Android-based, multi-sensor streaming and recording. We used the LSL framework in three mobile apps to send and record sensor signals. In three validation scenarios and a timing test (see Appendix A), we have shown results that suggest high reliability of our mobile LSL recording app, which is comparable to the desktop alternative LabRecorder. We argue that this report and the development of Record-a and Send-a can contribute to smartphone usage in health as well as research contexts [4] by providing a way to record, synchronize and stream sensor data between smartphones.

The King’s Fund report on digital health [4,40] identifies smartphones as being the most important contribution to personal health developments, even more important than areas such as genome sequencing or artificial intelligence [49]. In the context of smartphone usage, the authors explicitly identify sensor data collection on smartphones as being one of the most promising technologies for personal health monitoring applications - potentially replacing singular measurements at doctor’s appointments or during medical procedures (see also [50]). In both research contexts, as well as in personal health contexts, the collection of continuous, synchronized data can generate context to data that might otherwise be difficult to interpret.

Smartphones are ubiquitous and allow people to manage their personal health data (see, for example, [51,52,53]). A simple connection to cloud-based services to store and share data enables treatment over distance, which we facilitated by using the standardized LSL data handling framework. Data recorded and streamed within this open-source framework can be shared with remote healthcare workers, or alternatively, data can be processed directly on the device, providing the opportunity to detect patterns of interest quickly and to alert users. LSL can be added to most sensor hardware easily, highlighting the potential of our work.

Our results confirmed reliable and valid recordings in all validation scenarios. We did not observe data loss or technical issues. We evaluated our setup against the LabRecorder desktop version on Windows, to which we compared the resulting xdf files. In all scenarios, we found the xdf files contained almost identical content and therefore concluded that Record-a can reliably serve as an LSL recorder for Android devices. Taken together, we present a solution to synchronize sensor readings on off-the-shelf smartphones. Closing this gap contributes to establishing pocketable labs for everyone.

## Figures and Tables

**Figure 1 sensors-21-08135-f001:**
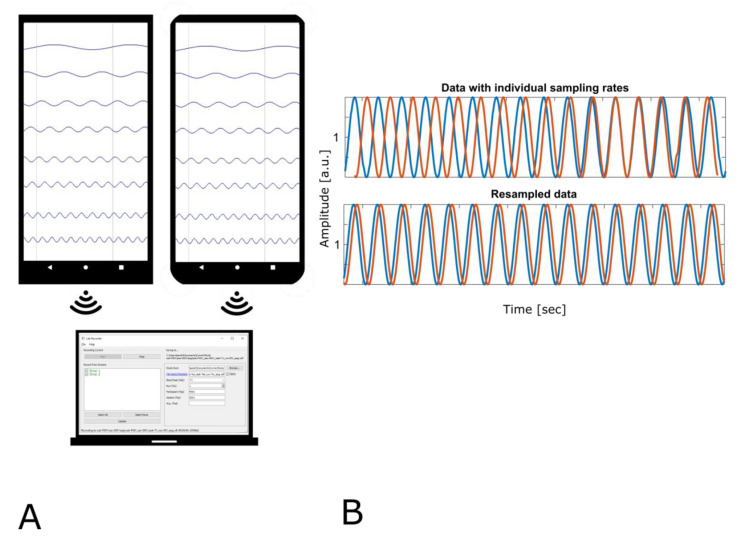
(**A**) Illustration of the setup of scenario 1A. Two phones were running the Sine-Wave app, while LabRecorder on PC recorded all data streams. Data were streamed with different sampling rates. (**B**) Illustration of a misalignment over time. Shown are 4 s of data. The upper plot shows data on their respective time scales diverging over time. The lower plot shows temporally aligned (resampled) data.

**Figure 2 sensors-21-08135-f002:**
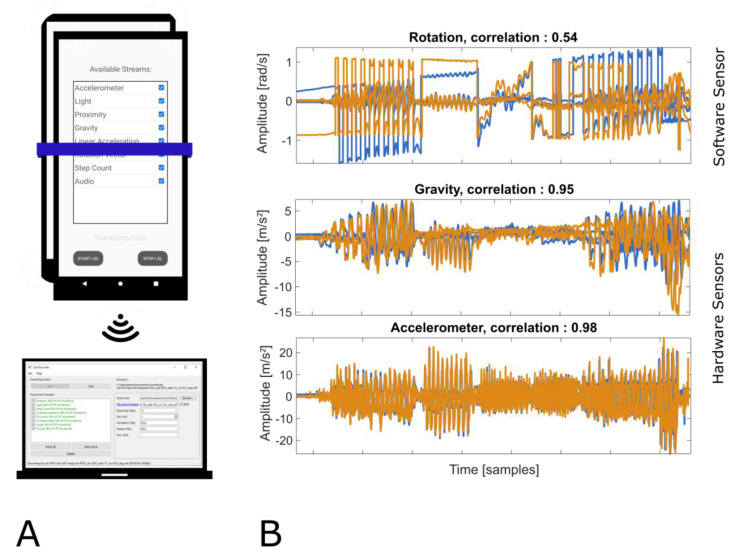
(**A**) Illustration of scenario 1B setup using sensor data as a data source. Two phones were moved together while Send-a was streaming sensor values. On PC, LabRecorder was recording all data streams to file. (**B**) Nine minutes of data recorded from three different sensors. Displayed is one channel for every sensor from phone 1 (blue) and phone 2 (orange).

**Figure 3 sensors-21-08135-f003:**
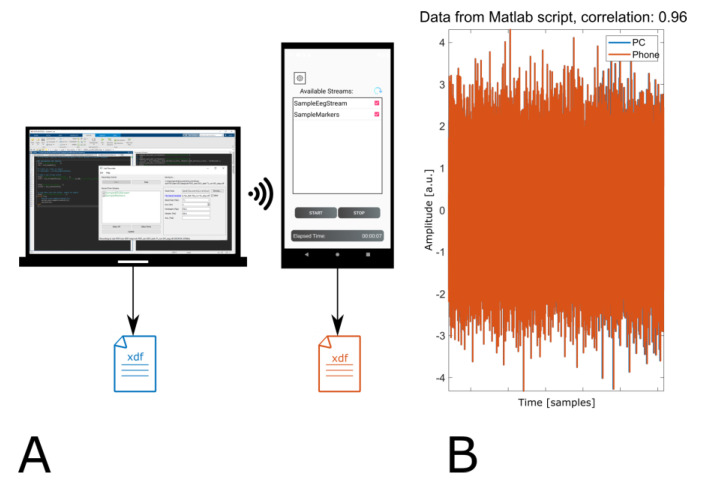
(**A**) Illustration of scenario 2. A PC streamed data which was recorded by Record-a on phone and LabRecorder on PC simultaneously. (**B**) 9.5 min of data from the same channel recorded on the PC and on the phone.

**Figure 4 sensors-21-08135-f004:**
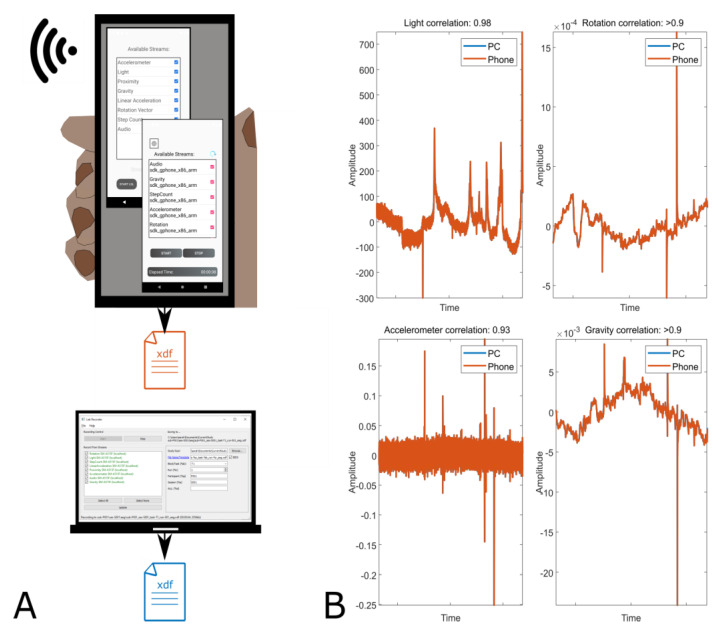
(**A**) Illustration of scenario 3. Both apps, Send-a and Record-a, were running on one phone, streaming and recording sensor data. On PC, LabRecorder was recording all data for validation purposes. (**B**) Selection of sensor data recorded on the PC and on the phone. Shown are 22 min of data from one channel per sensor.

## Data Availability

Data are available in this public repository: https://github.com/s4rify/Pocketable-Labs, accessed on 16 November 2021.

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
