# Peer review of "Pocketable Labs for Everyone: Synchronized Multi-Sensor Data Streaming and Recording on Smartphones with the Lab Streaming Layer"

_sensors, 2021, doi:10.3390/s21238135_

Round 1

Reviewer 1 Report

The paper is overall well written, describing three Android apps for data acquisition and their validation.  Although the apps could be useful in some applications, my feeling is that the paper is of limited interest to a scientific journal audience. 

Author Response

Reviewer 1:

The paper is overall well written, describing three Android apps for data acquisition and their validation.  Although the apps could be useful in some applications, my feeling is that the paper is of limited interest to a scientific journal audience.

Response:

We thank the reviewer for this important comment, which implies that we have not communicated the scientific value of our contribution in the original manuscript. The field of mHealth is currently transforming how health systems work and how care will be delivered in the future. Please note that this is not only our personal opinion, but the opinion expressed by experts in the field (e.g.: https://www.kingsfund.org.uk/publications/digital-revolution). Our paper contributes a very important tool, namely the possibility of time-synchronized recording of sensor streams. This possibility will be essential in translating basic research ideas into application. For example, modern neurorehabilitation efforts motor stroke rehabilitation use EEG neurofeedback training (e.g., https://journals.sagepub.com/doi/full/10.1177/1550059417717398) and these systems will clearly improve in performance if motion sensor data, such as signals from 9DOF IMU sensors, can be synchronized to EEG. Given that stroke is the world leading cause for adult disability, any effort in improving recovery and rehabilitation will eventually pay off.

In response to this comment, we have adapted the motivation in the introduction of the manuscript. In addition, we revised the discussion to emphasize the relevance of our findings for the respective fields. We have highlighted the corresponding sections in the manuscript.

Reviewer 2 Report

The discussion in this paper is not sufficient, and some descriptions may require more supporting information than simply comparing the similarity between the specific sine wave excitation signal with time analysis to confirm the results. Moreover, the functions and design of smartphones are a bit complicated, and it is not recommended to discuss certain brands and specific comparison cases. It should cover a wider range of parameters and the communication modules or architectures used to confirm their differences. If it is a signal comparison, adding more objective parameters or using a verified acoustic test system to produce a more specific report is recommended. In addition, it is also recommended that the length and description of the article should be more concise and specific.

Author Response

Reviewer 2:

Issue #1 “The discussion in this paper is not sufficient, and some descriptions may require more supporting information than simply comparing the similarity between the specific sine wave excitation signal with time analysis to confirm the results.”

Response: In response to this comment and in agreement with suggestions from reviewer 1 we have reworked the introduction and discussion sections. We are confident that the revised manuscript provides sufficient information for the reader to identify the relevance of our study and the validity of our findings.   

Issue #2 “Moreover, the functions and design of smartphones are a bit complicated, and it is not recommended to discuss certain brands and specific comparison cases. It should cover a wider range of parameters and the communication modules or architectures used to confirm their differences.”

Response: We agree with the reviewer but would like to point out that we cannot provide a systematic test of all major smartphone systems on the market. While we are confident that our apps work on many (if not most) devices, we want to make clear to the reader that some limitations may exist. Therefore, we consider it important to explicitly mention the smartphone systems we have used

In response to this comment, we have removed smartphone brand names where possible.

Issue #3 “If it is a signal comparison, adding more objective parameters or using a verified acoustic test system to produce a more specific report is recommended.”

We would like to note that the features we use for comparing the different conditions are objective measures and could be easily replicated independently, without the need for sophisticated test technology.

In response to this comment, we now state in more detail how performance metrics were obtained and evaluated. Please note that all timing tests and other evaluation scripts will be made freely available on our GitHub repository, which is linked in the paper. Thereby we can achieve maximum objectivity and reproducibility.

Issue #4 In addition, it is also recommended that the length and description of the article should be more concise and specific.

We agree and would like to note that complying to this issue has been guiding our revisions.

Round 2

Reviewer 1 Report

OK, now I understand the significance of the paper.

Reviewer 2 Report

OK, now I understand what is the possibility of time-synchronized recording of sensor streams. The contribution is good to implement in the mobility of the various smartphones for health care purposes.

BTW, the finished manuscript requires the format according to the Journal of MDPI sensor.